# The Relationship between Surface Roughness, Capillarity and Mineral Composition in Roofing Slates

**Víctor Cardenes** [1],*, **Alberto García** [2], **Eduardo Rodríguez** [2], **Antolín Hernández Battez** [2], **Santiago López-Piñeiro** [3], **Vicente G. Ruiz de Argandoña** [1] and **Álvaro Rubio-Ordoñez** [1]

[1] Geology Department, University of Oviedo, C/Jesús Arias de Velasco s/n, Oviedo, 33005 Asturias, Spain; vgargand@geol.uniovi.es (V.G.R.d.A.); rubioalvaro@uniovi.es (Á.R.-O.)

[2] Department of Construction and Manufacturing Engineering, University of Oviedo, 33005 Asturias, Spain; garciamaralberto@uniovi.es (A.G.); eduardo@uniovi.es (E.R.); aehernandez@uniovi.es (A.H.B.)

[3] Departament of Architectural, Civil and Aeronautical Buildings and Structures, University of Coruña, 15001 A Coruña, Spain; santiago.lopezp@udc.es

* Correspondence: cardenesvictor@uniovi.es

**Abstract:** Roofing slates are a category of building stones which have a very distinctive feature: High fissility, which allows them to be split into tiles that are thin, regular and large. There are several types of roofing slates, depending on their lithology. The four main lithologies are low-grade slates, slates stricto sensu, phyllites, and mica-schist. Occasionally, other rocks such as quartzites, serpentinites, or shales, can also be used as roofing slates. Roofing slates must ensure waterproofing, a quality that depends on both the rock and the installation. Installation must therefore take into account parameters such as the pitch, orientation, and overlap of the tiles in order to avoid capillarity, which could jeopardize waterproofing. These parameters are usually included in installation manuals. However, despite the fact that roughness is a parameter known to have an important effect on capillarity, it has never been thoroughly analyzed. Roughness varies depending on the type of roofing slate, but installation manuals do not take this factor into account. This study has measured surface roughness in different types of roofing slates using a laser scanner and determined the capillarity values along and across the grain direction. Furthermore, the role of dissolved salts in capillarity has likewise been studied.

**Keywords:** roofing slate; capillarity; salts; roof installation; surface roughness

## 1. Introduction

Roofing slates constitute a special range of metamorphic rocks that, as their name suggests, are typically used as shingles. Humanity has employed this type of rocks since prehistoric times. The main and necessary characteristic for a rock to be installed on a roof is its ability to be split into flat, thin, and durable shingles, and only a few rocks from across the earth's surface fulfill these special requirements.

It is important to make clear the distinction between two terms that can sometimes lead to misunderstanding: Roofing slate and slate. Roofing slate is any metamorphic rock that can be used to manufacture shingles, while slate is specifically a fine-grained rock from the green-schists facies which has developed a slaty cleavage [1]. While the most common rock used for roofing slate is slate s.s., several other rocks can be used as well, such as phyllite, mica-schist, and low-grade slates. Thus, according to the International Roofing Slate Classification (IRSC) [2], roofing slates are classified according to their petrology (low-grade slate, slate s.s., phyllite, and mica-schist) and color

(black-grey, purple-red, and green) into twelve lithotypes. Petrology gives general information about the mechanical performance of the slate shingle, while color gives information about the average mineralogy. Other roofing slate classifications have been proposed based on commercial qualities [3]. Although the manufacturing process is rather simple, the mining requirements are quite specific [4], meaning that roofing slate outcrops are rather scarce worldwide. In summary, not all slates can be used for roofing, and not all roofing slates are, in fact, slates. Shingles are split along this slaty cleavage, which is in fact the structure that defines this type of rocks. Slaty cleavage ($S_1$) defines a set of parallel flat surfaces formed during metamorphism. There is another important set of planes: The sedimentation ($S_0$). The intersection of these two planes generates a set of lines, called intersection lineation ($L_0$) (Figure 1A). Roofing slates are thus rocks with a very strong structural anisotropy, as a result of their low-grade metamorphism. This structural anisotropy is expressed by $S_1$, which in turn is responsible for the high fissility of these rocks. Roofing slate is also a durable and stable product, with better performance against weathering than other materials [5]. A recent study has shown that roofing slate is much less affected by moss growth than other covering materials [6].

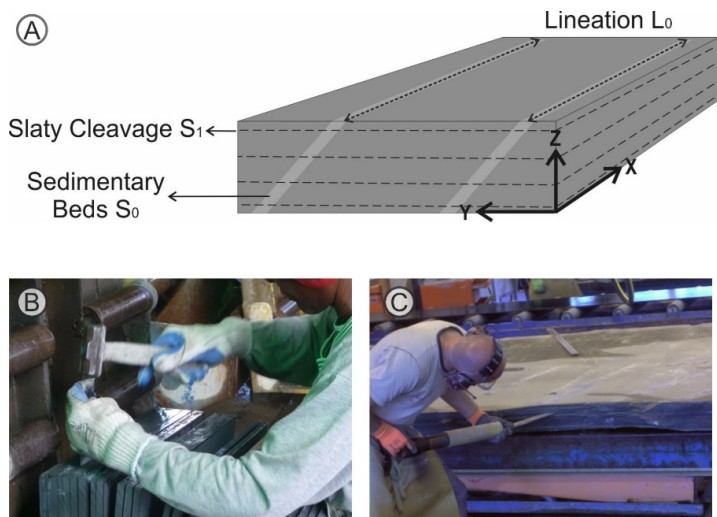

**Figure 1.** Relationships between sedimentation ($S_0$), slaty cleavage ($S_1$), and their intersection. (**A**) slate shingles are cut with their length parallel to lineation ($L_0$), in order to achieve optimal mechanical resistance; (**B**) splitting with a hammer and chisel in Minas Gerais, Brazil; and (**C**) splitting Alta quartzite with a pneumatic hammer in Alta, Norway.

This simple production process, mentioned above, involves the following steps. First, massive slate blocks are detached from the quarry (or mine) front, then sawed into smaller blocks, before being split using a hammer and chisel (Figure 1B,C). The resulting plates are then trimmed according to the shapes and dimensions required. Other natural stones may have different surface finishes (e.g., polished, flamed, and sawn), which each give the rock a different look [7]. Slate shingles, however, do not have any type of surface finish, just the surface as-is after splitting. The roughness of this surface depends on mineralogy, grain size, and, especially, on lineation, which creates a linear pattern on the surface. Some varieties of roofing slates are even commercially known by this pattern (Hebra and Grain varieties). The lineation is the most important structure of roofing slate shingles, since it defines the direction of the maximum mechanical resistance. Shingles have to be cut with their length parallel to the lineation in order to achieve this maximum mechanical resistance [8]. Because of this, slate roofs usually have all the shingles installed with this lineation vertically. This lineation is sometimes clearly distinguishable on hand specimens, while other times is not visible at all. In any case, during the manufacturing process lineation is usually identified and followed when sawing the roofing slate blocks.

Surface roughness is strongly influenced by the measurement scale [9,10]. Surface parameters can be grouped into amplitude parameters, spacing parameters, and hybrid parameters [11]. Amplitude

parameters measure vertical features of the surface, while spacing parameters describe horizontal features. Finally, hybrid parameters are a combination of amplitude and spacing, such as slope or curvature. Roughness is also one of the determining factors for capillary action, which allows liquids to flow along a surface against the pull of gravity [12,13]. This effect takes place when the intermolecular strength of the liquid is lower than the adhesion forces of the liquid with the surface. The liquid moves up until its surface tension is balanced with the weight of the liquid attached.

As pointed out before, roof design is greatly conditioned by the capillarity of the slate surfaces, which may jeopardize the waterproofing of the roof. A slate roof is a rather simple structure, composed of two well-defined parts: The supporting bed (the frame to which the slate shingles are attached) and the covering elements (slate shingles) (Figure 2A,B). Values for this overlapping are compiled in tables available in every roofing manual. However, capillarity still occurs, sometimes due to a defective installation, but other times due to a miscalculation of the capillary effect. Traditional roofing slate methods can provide the same waterproofing as any other type of modern method [14].

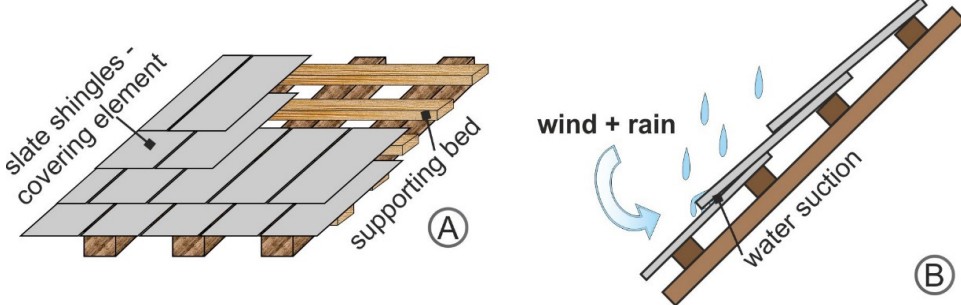

**Figure 2.** (**A**) diagram of a typical slate roof. (**B**) rainwater, propelled by the wind, penetrates between slate shingles.

The adverse effect of capillarity is controlled by both the overlapping of the slate shingles and the pitch [15,16]. Rain intensity is the third major factor influencing capillarity, but, unlike the previous two, rain cannot be controlled, only taken into account. According to the UK standard BS 5534 Code of practice for slating and tiling [17], the minimum pitch should not be lower than 20°, while the Spanish UNE 22190 Construcción de cubiertas inclinadas y revestimiento de paramentos verticales con pizarra [18] recommends the pitch be higher than 16.7°. This last standard also takes into account the effect on capillarity of the different attachment systems. Roofing slate shingles can be attached using nails or hooks, depending on the architectural style. Nail attaching is the more traditional method, while hooks have only been used since modern times. While the nails were originally made of wood, nowadays they are copper or galvanized steel, while the hooks are mostly galvanized steel. Nails are typical in Central European countries, the UK, and the US, while hooks are mainly used in Spain. There are two main differences between these two systems, one aesthetic and one functional. Regarding aesthetics, hooks are visible while nails are hidden by the overlying slate. In terms of functionality, is very easy to replace a broken shingle in a hooked roof, as one need only bend the hook head with a clamp, substitute the shingle, and return the hook head to its initial position. On the other hand, to replace a broken shingle in a nailed roof one must pull out the nail with a special tool, substitute the broken piece, and then use a hook to attach the new shingle. Furthermore, hooks secure the shingles more securely than nails do, and better withstand the wind. Nails and hooks apply the maximum attaching force at different points in the shingle, which means that the shape of the maximum capillarity ascension is also determined by the attachment system (Figure 3), due to this difference in the attaching forces [19]. However, the maximum height reached by the water does not depend on the attachment system, but rather on the physical interactions between the roofing slate surface, liquid, and wind force.

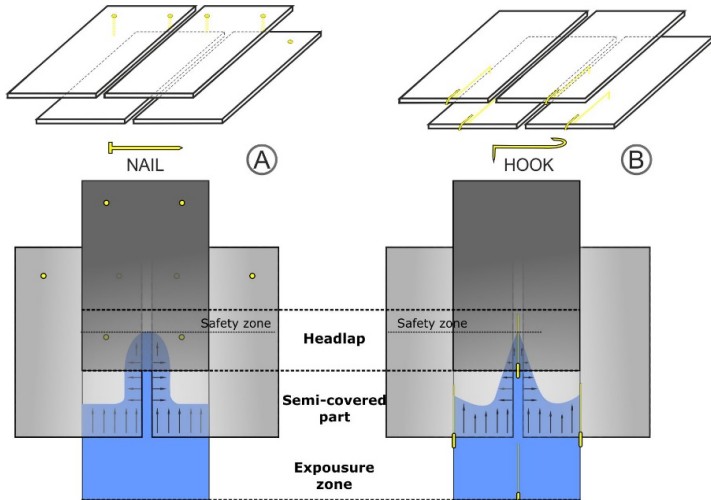

**Figure 3.** Attachment by nail (**A**) and hook (**B**), with their respective capillarity shapes.

Dissolved salts can also play an important role in capillarity. The height a liquid can reach by capillarity on a flat surface is determined by the surface tension, viscosity, and density of the liquid. Because dissolved salts modify the viscosity of water, this is also a factor that affects ascension. The addition of salts to the liquid (distilled water in this case) changes these values, generally increasing them. Guimaraes et al. [20] found that, after several cycles of saline solution saturation, capillarity decreased as compared with distilled water. Salts may also threaten the structural integrity of the rock due to a crystallization–volume increase process. This fact is governed by the porosity system of the rock. Roofing slate has very low porosity values, as can be deduced by the water absorption test results. Besides, roofing slate shingles are not thick enough to be affected by salt disintegration. An analysis of more than 150 technical sheets provided by companies has yielded an average value of 0.28%, with a standard deviation of 0.11 [1]. However, under special climatic conditions, salt crystallization may act as a weathering mechanism for slate bricks or slabs [21], in the same way that freeze-thaw does. However, while freeze-thaw is compiled in the European Norm for roofing slates [22], salt crystallization is not, despite both mechanisms being controlled by porosity.

There are many different roofing slate varieties, and some have very different surface roughness (and hence capillarity) values from the ones used to calculate overlapping values in the tables from installation manuals. The present paper therefore seeks to measure the roughness of several roofing slate varieties, using different saline solutions, and then compare the results with the capillarity values determined in the directions parallel and perpendicular to the lineation $L_0$. Our results show that surface roughness should be carefully taken into account when installing a slate roof. The overlap of the shingles can be adjusted to optimize the weight and economic cost of the roof using obtained data, which are not included in any installation manual.

## 2. Materials and Methods

Sixteen types of roofing slates were selected for this study (Table 1). Samples were chosen to cover the range of different surfaces and textures found on the market. However, not all varieties have the same representation on the market. As can be seen in Table 1, there are certain samples (05 and RIM) which were selected based on their particular surface features, despite being unavailable on today's market. Sample 05 is a unique slate shingle with a very marked lineation, not really suitable for roofing, while RIM belongs to the exhausted outcrop of Rimognes, in France, and is also a type of roofing slate unique, due to the occurrence of magnetite crystals on its surface. Some samples clearly show the lineation, while other not. For these last samples, we have assumed that they have been manufactured with the lineation parallel to the length of the shingle, as it should be to obtain the

best mechanical resistance. In most of the cases, the samples were obtained directly at the quarry. Each sample consisted of two shingles with similar dimensions.

Surface roughness was determined using a Leica DCM 3D confocal microscope. A 41 by 40 mm$^2$ rectangular area of each roofing slate was scanned using a 5× objective magnification. This area was obtained by stitching together 520 single confocal measures, with 15% overlap. The data obtained by confocal microscope was post-processed using a data processing software (Matlab©) (v. R2020a, The MathWorks Inc., Natick, MA, USA) to determine the surface roughness parameters: Arithmetic height (Sa), root mean square (Sq), skewness (Ssk), and kurtosis (Sku), according to the ISO 25178-2 standard [23]. Sa gives an overview of the surface roughness, the higher the value the rougher the surface. The other parameters give information about the distribution and shape of this roughness.

**Table 1.** Roofing slate samples. "Code/IRSC" refers to the lithotype according to the International Roofing Slate Classification. The letter is for color family (B: Black-grey; G: Green; and R: Purple-red), while the number is for rock type (0: Low-grade slate; 1: Slate; and 2: Phyllite). ST indicates other types of rock used as roofing slate. In this case, ALT is a quartzite, and SPT a serpentinite. "Market" represents the availability on the market. "High" indicates widely available roofing slates; "Medium" roofing slates which are available but scarce; "Low" scarce and special roofing slates that are only found in a restricted area; and "No market" is for roofing slates which are not sold at all. For the Surface Texture columns, blank space means no present, X low occurrence, XX medium occurrence and XXX high occurrence.

| CODE | IRSC | Market | Location | Surface Texture | | | | |
|------|------|--------|----------|--------|---------|--------|----------|----------|
| | | | | Smooth | Grained | Flaked | Lineation | Features |
| BRA | B0 | Low | Minas Gerais, Brazil | X | X | | | |
| WAM | B0 | Low | Gauteng, South Africa | | | X | | |
| BUR | B0 | Low | Lake District, UK | | X | | X | |
| 01 | B1 | High | Valdeorras, Spain | | | XX | | |
| 02 | B1 | High | Valdeorras, Spain | X | | | X | |
| 03 | B1 | High | Valdeorras, Spain | X | | | XX | |
| 04 | B1 | High | Valdeorras, Spain | X | | | | |
| 05 | B1 | No market | Valdeorras, Spain | X | | | XXX | |
| 06 | B1 | Low | Valdeorras, Spain | X | | | XX | Pyrite |
| PIV | B1 | High | Valdeorras, Spain | X | | | X | |
| PEN | R1 | Medium | Penrhyn, UK | X | | | | |
| NYR | R1 | Low | New York, USA | X | | | | |
| RIM | G1 | No market | Rimogne, France | | X | | X | Magnetite |
| OSO | G2 | Medium | Lugo, Spain | | | X | | |
| SPT | ST | Low | Valmalenco, Italy | | X | X | XXX | |
| ALT | ST | Low | Alta, Norway | | X | | | |

Capillarity was measured by joining together two samples from each roofing slate variety using elastic bands, and immersing them in distilled water, in a NaCl solution, and in a Na$_2$SO$_4$ solution, for 24 h at room temperature. The concentration of the saline solutions was 3.5% for NaCl (the average content in sea water) and 14% for Na$_2$SO$_4$ (as in EN 12370 Natural Stone Methods. Determination of resistance to salt crystallization, [24]). At this temperature, and for the given concentrations, NaCl has a viscosity of 0.950 mPa·s and a density of 1.023 g·cm$^{-3}$, while for Na$_2$SO$_4$ the values are 1.545 mPa·s and 1.055 g·cm$^{-3}$. These are the two most commonly found salts in natural environments, and are responsible for most rock weathering caused by salt crystallization, including slate slab deterioration [25]. Capillary ascension was measured in two directions: Parallel and perpendicular to

lineation. Likewise, the average and maximum values were also measured. For some the samples, the lineation was clearly visible, but for others this was less obvious. Nevertheless, we know that the length of the shingles coincides with the lineation, so in doubtful samples, we took length as lineation. The average and maximum capillary ascension was measured for each sample, placing them at 90° with respect to the bottom of the tray. As pointed out before, pitch angle should be higher than 20° (16.7° for the Spanish standard). The capillary ascension values obtained with this 90° angle can be converted to other angles using a basic trigonometric conversion (Figure 4), were H (capillary) = c (water ascension) for 90°. Pitch angle is decided when calculating the roof, taking into account both aesthetics and functional design. This paper provides capillary ascension values for 90°. However, values for other pitches can be determined by applying the formula in Figure 4, using the values found in this paper for 90°.

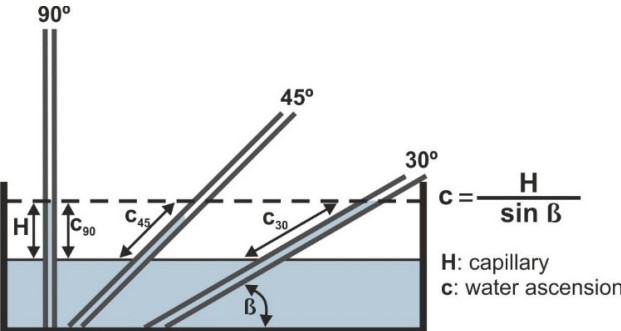

**Figure 4.** Capillary ascension depending on surface angle, and the trigonometric conversion used in this study.

Mineral determination and semi-quantification were conducted at the University of Oviedo's Scientific Services Department, using a Philips PW 1830 diffractometer, with a Cu cathode and a wavelength of $K\alpha = 1.5405$ Å. The angular scan was recorded from 2° to 60° $2\theta$. Semiquantification of the mineral phases was performed using the software XPowder (v. 2004.04.44 PRO, Universidad de Granada, Spain), using the Rietveld method, which has shown great accuracy in roofing slates [26]. Finally, in order to check for possible relationships between the measured parameters, a correlation matrix was built with the values of the average capillarity, mineral composition, and roughness, using the statistics suite SPSS v 15.0 for Microsoft Windows©.

## 3. Results and Discussion

The mineral composition (Table 2) was as expected for this type of rocks: The main minerals were quartz, chlorites, and mica, occasionally with some accessories such as pyrite and carbonates. Since roofing slates come from the metamorphism of pelitic sediments, the mineral composition is rather similar, regardless of age or location. Sample SPT has a different mineral composition, since this rock is a serpentinite, hence a rock with a rather different metamorphic history.

The study of surface roughness revealed that most of the samples presented Sa values below 150 (Table 2). Only four samples had values above 150 μm: 01, 05, ALT, and SPT. The first two are slates s.s. 01 has a very rough and flaky texture which conceals the lineation, while 05 is one of the exceptional samples chosen based on its texture, in this case with a very strong lineation that even causes the shingle to bend. ALT and SPT are roofing stones, ALT presents a significantly smaller number of phyllosilicates and more hard minerals (quartz and feldspar) than roofing slates, while SPT is wholly composed by phyllosilicates. Figure 5 shows a macro image and the confocal scanned topography of some of the samples, using the same z-axis limits (color scale). Samples with the greatest Sa presented different topographical properties.

**Table 2.** Results for the determination of roughness (RUG), capillary ascension, and mineralogy. Mineral abbreviations are Q: Quartz, Chl: Chlorites, Fs: Feldspars, Cte: Carbonates, Acc: Accessory minerals, Ant: Antigorite, Chy: Chrysotile, Srp: Serpentinite, and Liz: Lizardite. Directions parallel and perpendicular to $L_0$ are indicated as // and ⊥, respectively. Wt indicates water capillary ascension in cm. Values in parentheses in the capillary results refer to the maximum height reached, as in Figure 7C,D.

| CODE | Mineralogy (%) | | | | | | RUG Sa (μm) | Capillary Ascension (cm) | | | | | |
|---|---|---|---|---|---|---|---|---|---|---|---|---|---|
| | Q | Chl | Mica | Fs | Cte | Acc | | Wt // | Wt ⊥ | Na$_2$SO$_4$ // | Na$_2$SO$_4$ ⊥ | NaCl // | NaCl ⊥ |
| BRA | 65.5 | 9.1 | 6.2 | 19.2 | 0.0 | 0.0 | 110.7 | 4.8(6.1) | 5.2(5.6) | 4.5(6.4) | 4.5(6.0) | 4.6(5.2) | 4.7(–) |
| WAM | 49.5 | 4.1 | 30.6 | 5.7 | 9.7 | 0.4 | 136.0 | 4.8(7.0) | 5.2(6.0) | 3.8(6.0) | 5.5(7.5) | 3.8(–) | 4.7(–) |
| BUR | 64.1 | 18.2 | 17.6 | 0.0 | 0.0 | 0.0 | 114.5 | 5.2(5.8) | 5.2(5.3) | 3.6(–) | 3.6(4.4) | 4.5(5.9) | 4.3(–) |
| 01 | 47.7 | 11.0 | 22.5 | 17.9 | 0.0 | 0.9 | 173.3 | 4.8(5.8) | 5.1(–) | 4.8(5.8) | 4.7(5.7) | 4.4(–) | 4.2(5.5) |
| 02 | 33.5 | 13.4 | 39.7 | 13.4 | 0.0 | 0.0 | 51.7 | 4.0(8.0) | 3.9(8.3) | 3.3(4.5) | 4.5(8.0) | 3.2(5.8) | 3.8(8.5) |
| 03 | 31.3 | 21.4 | 17.6 | 29.2 | 0.0 | 0.5 | 100.1 | 4.9(5.0) | 5.2(–) | 3.4(–) | 4.9(–) | 3.8(5.0) | 3.8(5.5) |
| 04 | 45.7 | 15.0 | 20.4 | 18.3 | 0.0 | 0.5 | 82.5 | 3.9(4.5) | 4.5(4.6) | 2.9(–) | 4.2(–) | 3.5(–) | 4.2(–) |
| 05 | 34.9 | 26.1 | 17.0 | 21.3 | 0.0 | 0.7 | 292.6 | 3.8(6.8) | 4.1(4.6) | 3.3(4.3) | 5.0(7.0) | 5.5(6.5) | 4.5(6.5) |
| 06 | 32.2 | 21.5 | 18.0 | 27.9 | 0.0 | 0.4 | 63. 0 | 3.3(4.5) | 4.4(6.0) | 3.0(4.4) | 3.8(–) | 3.6(–) | 3.8(5.4) |
| PIV | 41.6 | 9.9 | 15.8 | 31.6 | 0.0 | 1.1 | 94.8 | 3.3(5.5) | 4.9(–) | 2.8(–) | 3.6(–) | 3.5(4.5) | 4.2(–) |
| PEN | 47.5 | 14.9 | 11.0 | 19.3 | 0.0 | 7.3 | 126.0 | 4.2(5.0) | 5.2(6.2) | 3.0(3.0) | 5.5(5.9) | 5.0(6.0) | 4.5(6.5) |
| NYR | 37.3 | 1.6 | 9.8 | 12.9 | 30.9 | 7.6 | 95.6 | 4.5(5.0) | 4.9(5.8) | 3.4(3.6) | 4.2(4.5) | 4.0(5.7) | 3.8(4.4) |
| RIM | 35.3 | 6.6 | 30.4 | 27.7 | 0.0 | 0.0 | 127.7 | 4.8(7.0) | 5.0(6.0) | 4.8(5.8) | 5.5(7.5) | 4.5(8.0) | 5.0(9.3) |
| OSO | 35.5 | 10.0 | 39.0 | 15.6 | 0.0 | 0.0 | 117.3 | 4.6(5.0) | 5.3(6.9) | 3.5(4.5) | 4.5(6.0) | 4.2(5.1) | 4.1(–) |
| ALT | 58.9 | 0.0 | 21.6 | 19.5 | 0.0 | 0.0 | 401.1 | 5.7(–) | 6.2(–) | 5.0(5.0) | 7.0(9.0) | 4.2(5.5) | 9.8(14.8) |
| | Ant | Chy | Srp | Liz | | | | | | | | | |
| SPT | 26.4 | 37.5 | 22.9 | 13.2 | | | 202.1 | 5.4(6.5) | 5.9(6.9) | 4.9(6.5) | 5.5(8.0) | 6.2(9.5) | 5.5(6.3) |

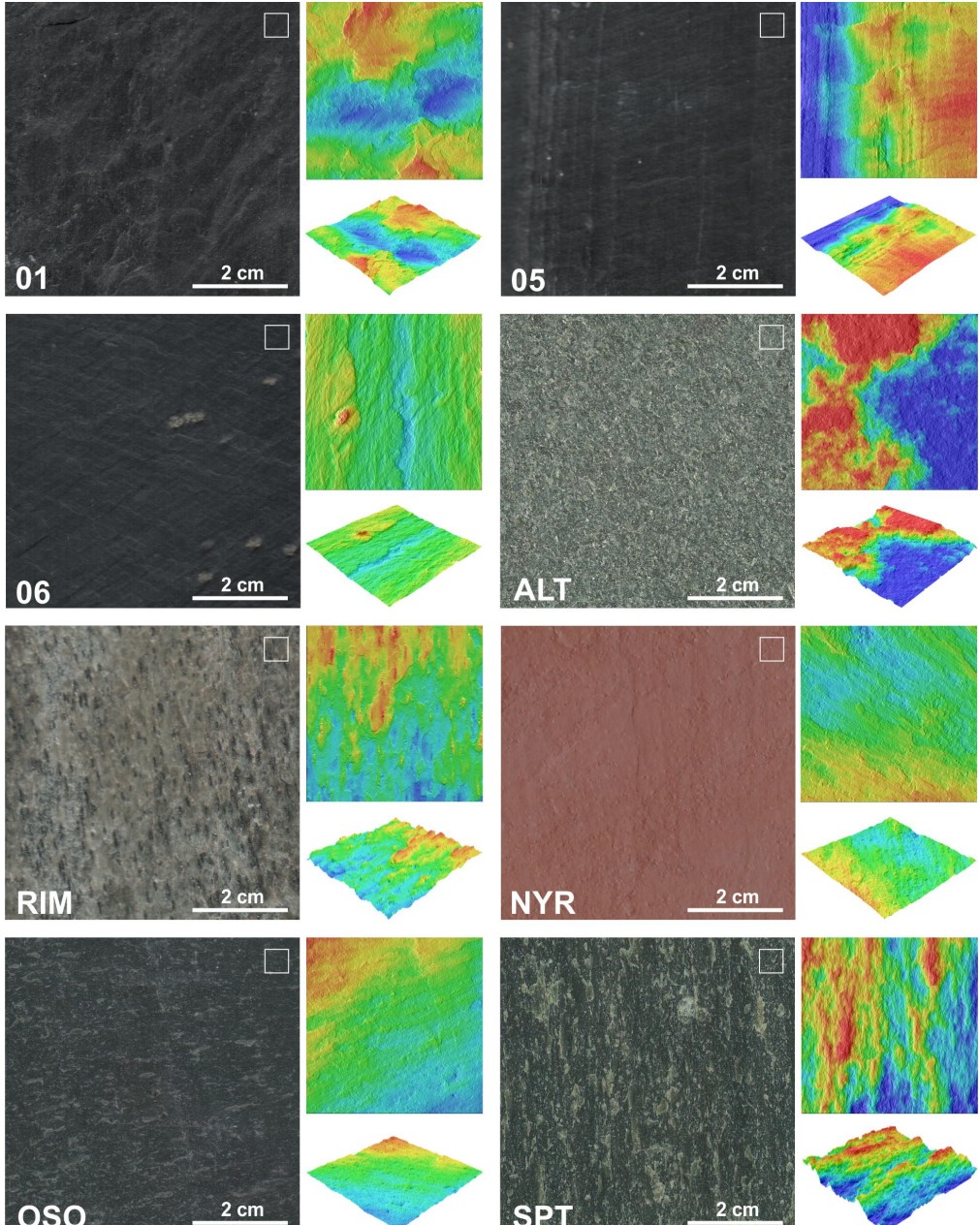

**Figure 5.** Macro aspect and laser roughness scan of some of the samples. The small white square in the upper right corner of each macro image represents the area scanned in the surface-roughness laser measurement. The vertical scale is the same for all the samples in the roughness renderings (between −500 μm (blue) and +500 μm (red)). Samples 01, 05, ALT, and SPT have greater differences in height than 06, RIM, NYR, and OSO.

The skewness analysis (Figure 6A) indicated that samples 05, BRA, and WAM had significantly negative values, which indicates that roughness values are caused mainly by valleys on the surface (negative z values). On the other hand, samples PEN and 03 exhibited significantly positive Ssk values, indicating the presence of crests (positive z values) on the surface. Kurtosis (Figure 6B) showed that slate 05 exhibited a sharp topography (Sku > 3), related to the sample's pronounced lineation, while sample ALT had a smooth surface (Sku < 3). The granulated surface of sample ALT resulted in a homogeneous height distribution of peaks and valleys, and as a result, Sku ≈ 2.

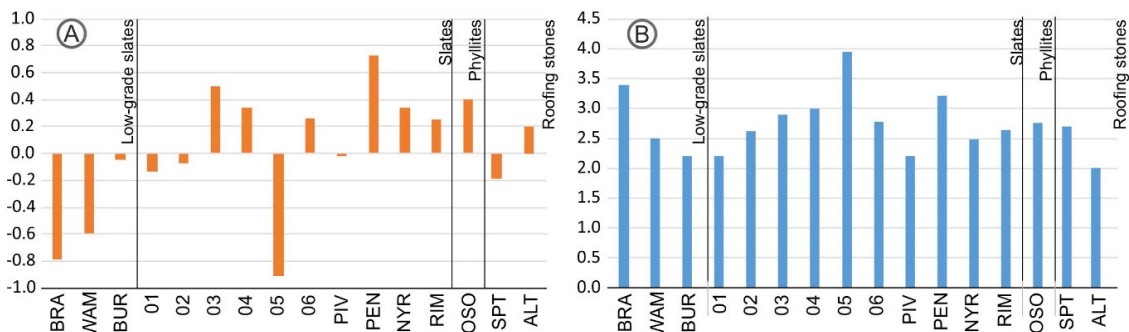

**Figure 6.** Skewness (**A**) and kurtosis (**B**) analysis of the height distribution for each sample. Skewness (**A**) represents the degree of symmetry of the surface heights across the mean plane: Positive values indicate a predominance of crests, and negative values indicate a predominance of troughs. Kurtosis (**B**) indicates the presence of significant high peaks and/or deep valleys (Sku > 3), or the absence thereof (Sku < 3) (i.e., smooth surfaces).

On the other hand, the results for capillarity (Figure 7, Table 2) have shown that the capillary values of the direction parallel to lineation are lower than those perpendicular to lineation for all three solutions. Distilled water exhibits higher values (an average of 4.5 cm parallel to lineation and 5.0 cm perpendicular to lineation), while $Na_2SO_4$ and NaCl have averages of 3.8 and 4.3 cm parallel to lineation, and 4.8 and 4.6 cm perpendicular to lineation, respectively. These results confirm the importance of manufacturing roofing slates with the length parallel to lineation, not only to achieve the optimum mechanical performance but also to minimize capillarity.

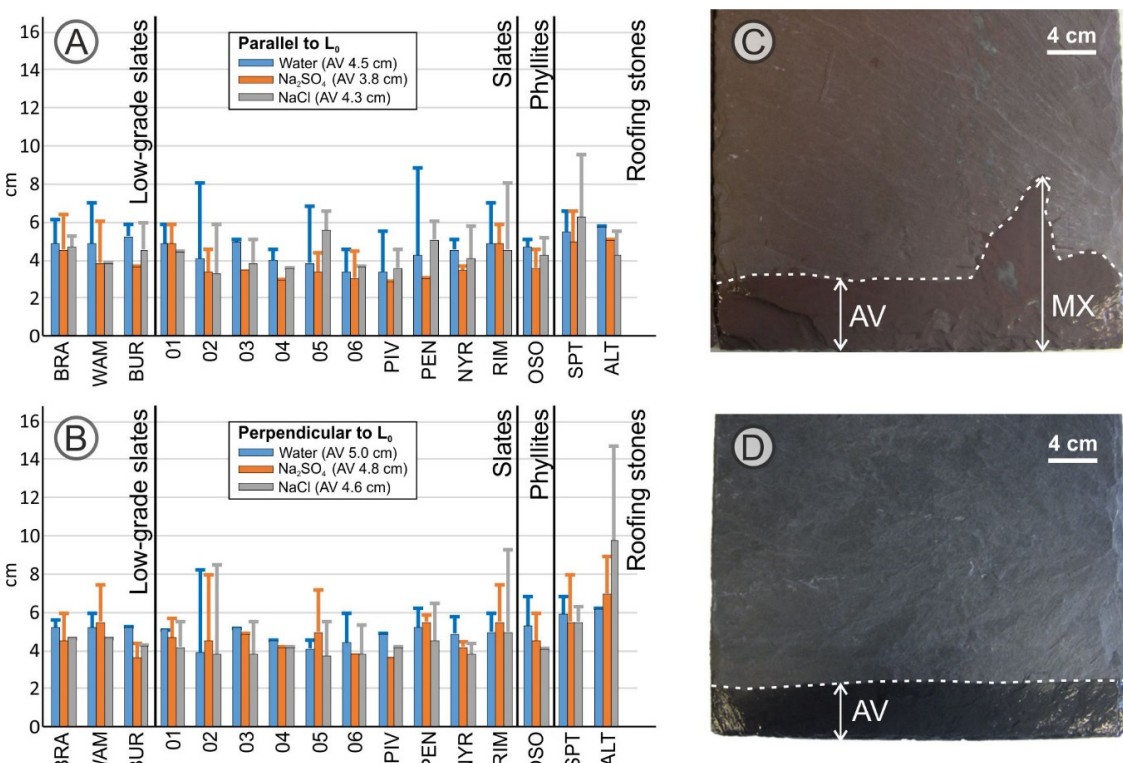

**Figure 7.** Capillary ascension for distilled water, $Na_2SO_4$ solution and NaCl solution. (**A**,**B**) show the average values (AV) for each solution are indicated in the legend. The maximum values are indicated by error bars. The samples are sorted by lithotype. (**C**,**D**) are examples of water capillary ascension on sample PEN and 01, showing the average and maximum (MX) values marked.

The correlation analysis (Table 3) between the parameters measured (capillarity, roughness, and mineral composition) detected the highest correlations between roughness and capillarity for saline solutions. Another set of correlations was also found between quartz/chlorites and capillarity, although the values are barely significant. Kurtosis and Skewness have not presented significant correlations with capillarity for any of the solutions in both directions. Regarding lithotype, with the increase of metamorphism, roofing slates usually tend to develop a visible lineation, which affects the surface roughness. Then, capillary values parallel to lineation are rather similar for all the samples, but capillarity perpendicular to lineation are higher for the samples with higher metamorphic degree (Figure 7).

**Table 3.** Pearson correlation for the results from Table 2. Sa is taken as the value for the arithmetic height. Correlations significant to 0.01 are marked **, while correlations significant to 0.05 are marked *. Correlations between mineral components have not been included, since they are not relevant to this study. Sample SPT has not been included for the mineralogical correlation due to its special mineralogy (Table 2).

| Correlation Matrix | Sa | Wt // | Wt ⊥ | Na$_2$SO$_4$ // | Na$_2$SO$_4$ ⊥ | NaCl // | NaCl ⊥ | Q | Chl | Mica | Fs | Cte | Acc |
|---|---|---|---|---|---|---|---|---|---|---|---|---|---|
| **Sa** | 1 (16) | 0.479 (16) | 0.487 (16) | 0.743 (**) (16) | 0.539 (*) (16) | 0.766 (**) (16) | 0.493 (16) | 0.297 (15) | −0.189 (15) | −0.064 (15) | 0.007 (15) | −0.131 (15) | −0.129 (15) |
| **Wt //** | | 1 (16) | 0.795 (**) (16) | 0.597 (*) (16) | 0.787 (**) (16) | 0.603 (*) (16) | 0.403 (16) | 0.543 (*) (15) | -0.468 (15) | 0.077 (15) | −0.429 (15) | 0.069 (15) | −0.126 (15) |
| **Wt ⊥** | | | 1 (16) | 0.564 (*) (16) | 0.635 (**) (16) | 0.707 (**) (16) | 0.384 (16) | 0.561 (*) (15) | −0.571 (*) (15) | −0.145 (15) | −0.099 (15) | 0.012 (15) | 0.033 (15) |
| **Na2SO4 //** | | | | 1 (16) | 0.613 (*) (16) | 0.765 (**) (16) | 0.375 (16) | 0.156 (15) | −0.409 (15) | 0.164 (15) | 0.045 (15) | −0.088 (15) | 0.006 (15) |
| **Na2SO4 ⊥** | | | | | 1 (16) | 0.612 (*) (16) | 0.460 (16) | 0.433 (15) | −0.522 (*) (15) | 0.111 (15) | −0.076 (15) | −0.085 (15) | −0.302 (15) |
| **NaCl //** | | | | | | 1 (16) | 0.161 (16) | 0.478 (15) | −0.552 (*) (15) | 0.019 (15) | −0.002 (15) | −0.132 (15) | −0.162 (15) |
| **NaCl ⊥** | | | | | | | 1 (16) | 0.262 (15) | 0.205 (15) | −0.349 (15) | −0.078 (15) | −0.115 (15) | 0.210 (15) |

## 4. Conclusions

The standard procedure for manufacturing roofing slate tiles using the lineation as the length of the shingle, has proven to be important not just in terms of achieving optimal mechanical resistance, but also for diminishing capillarity. Our results show that capillarity is always lower in the direction parallel to lineation, between 90% and 80% of the value in the direction perpendicular to lineation.

Roughness is positively correlated with capillarity, especially for saline solutions in the direction parallel to lineation (0.766 for NaCl and 0.743 for Na$_2$SO$_4$, values for Pearson correlation coefficient). This means that for environments where saline sprays occur, rougher roofing slate surfaces have a greater risk of capillarity.

The mineral percentage of quartz has a slightly positive correlation with distilled water capillarity, while the percentage of chlorites exhibits a slightly negative correlation with salt capillarity. On the other hand, quartz and chlorites have no correlation at all with roughness. Roughness seems to be the result of tectonic lineation rather than mineral composition.

**Author Contributions:** Conceptualization, V.C., A.G. and E.R.; methodology, V.C., A.G., E.R. and A.H.B.; software, V.C., A.G. and E.R.; validation, S.L.-P. and A.R.; formal analysis, A.G., E.R. and Á.R.-O.; investigation, V.C.; resources, A.R., V.G. and S.L.-P.; writing—original draft preparation, V.C.; writing—review and editing, V.C., A.G., E.R., A.H.B., V.G.R.d.A.; visualization, V.C.; supervision, V.C.; funding acquisition, Á.R.-O. and V.G.R.d.A. All authors have read and agreed to the published version of the manuscript.

**Funding:** Víctor Cárdenes is grateful to his grant PA-18-ACB17-11, from the Program Marie-Curie COFUND funded by the European Union, Government of Asturias (Spain) and the Spanish FICYT.

**Conflicts of Interest:** The authors declare no conflict of interest. The funders had no role in the design of the study; in the collection, analyses, or interpretation of data; in the writing of the manuscript, or in the decision to publish the results.

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
