# Peer review of "The Relationship between Surface Roughness, Capillarity and Mineral Composition in Roofing Slates"

_minerals, doi:10.3390/min10060539_

Round 1

Reviewer 1 Report

This article focuses on the problems of the shingles making up roofing slates diffusely used as the in ancient buildings. Particularly are studied the relationship between surface roughness of materials, capillarity and mineral composition. The obtained results suggest useful information for optimal manufacturing of the roofing slates and are very interesting. Anyway, it is necessary to carry out accurate checks on some parts of texts/figures/tables to make corrections and/or improvements that I have indicated as follows.

Row 86 – (Figure 2) please cite Figure 2A and 2B in the text.

Row 142 – I Think is better to specify the number of samples analyzed (16 samples) in the text.

Row 163 – “3: mica-schist”. This type is not present in the names of samples included in Table 2 therefore I think is better to delete it.

Row 171 - Please insert the norm you have cited (EN 12370 Natural Stone Methods. Determination of resistance to salt crystallization) in the References. Check also the reference n.20 not present in the text.

Row 193 – “wavelength of Κα=1.5405(1) substitute with (Å). The angular scan was detected from 2 to 15° 2θ.” It is an oversight; I think it used an angular scan from 3-4° to 60-70° 2θ. Authors can also improve this part adding other information as the software used for the qualitative and quantitative analyses and the scan step.

Row 199 – “The mineral composition (Table 2) was as expected for this type of rocks: the main minerals were quartz, chlorites and mica, with some accessories such as pyrite and carbonates”. I understand that is a general framework of the mineralogical composition of the samples. But it needs to be clarified because carbonates are present only in two samples and not as accessory minerals especially in sample NYR.

Row 204 – It is better to cite sample ALT as the former and SPT sample as the latter as reported in Table 2.

Rows 207-208 – “SPT and ALT are roofing stones, with a significantly smaller number (is better with smaller amount or quantity) of phyllosilicates and more hard minerals (quartz and feldspar) than roofing slates. But SPT sample is wholly made up by phyllosilicates.

Rows 216 – “The skewness analysis (Figure 6) indicated that…….” you cited Figure 6B therefore I suggest shifting it in the left part of Figure 6 rename as Figure 6A.

Rows 219 – 221. Please cite Figure 6A in the text now renamed as 6B.

Rows 226-227 – “Distilled water exhibits higher values (an average of 4.5 cm parallel to lineation and 5.0 cm perpendicular to lineation), while Na2SO4 and NaCl have averages of 3.8 and 4.8 cm parallel to lineation, and 4.3 and 4.6 cm perpendicular to lineation, respectively.” You inverted some values please check accurately. I think that you also inverted the values in Table 2. I understand the symbols Wt but it is better to specify in table caption. Table 2 could be reorganized in the same order in which you have discussed the data in the text shifting the mineralogy on the left part of the table, then the surface roughness values and finally on the right part of Table 2, the capillary ascension data.

Row 250 – In Figure 6A and B substitute the colour of symbol NaCl with grey as indicated in Figure A and B.

Figure 5 – It is better to insert a scale bar near the macro images.

Refences – Check accurately and where possible, use Journal title abbreviations.

Author Response

Row 86 – (Figure 2) please cite Figure 2A and 2B in the text.

Done

Row 142 – I Think is better to specify the number of samples analyzed (16 samples) in the text.

Done

Row 163 – “3: mica-schist”. This type is not present in the names of samples included in Table 2 therefore I think is better to delete it.

Done

Row 171 - Please insert the norm you have cited (EN 12370 Natural Stone Methods. Determination of resistance to salt crystallization) in the References. Check also the reference n.20 not present in the text.

Reference to EN 12370 included. Reference [20] to EN 12326 is included in the text, please see row 132

Row 193 – “wavelength of Κα=1.5405(1) substitute with (Å). The angular scan was detected from 2 to 15° 2θ.” It is an oversight; I think it used an angular scan from 3-4° to 60-70° 2θ. Authors can also improve this part adding other information as the software used for the qualitative and quantitative analyses and the scan step.

We have corrected these mistakes referring the XRD, and also added the reference to the XRD software.

Row 199 – “The mineral composition (Table 2) was as expected for this type of rocks: the main minerals were quartz, chlorites and mica, with some accessories such as pyrite and carbonates”. I understand that is a general framework of the mineralogical composition of the samples. But it needs to be clarified because carbonates are present only in two samples and not as accessory minerals especially in sample NYR.

Indeed the role of accessory minerals need to be clarified. We have modified the discussion as: the main minerals were quartz, chlorites and mica, with occasionally some accessories such as pyrite and carbonates.”

Row 204 – It is better to cite sample ALT as the former and SPT sample as the latter as reported in Table 2.

Done

Rows 207-208 – “SPT and ALT are roofing stones, with a significantly smaller number (is better with smaller amount or quantity) of phyllosilicates and more hard minerals (quartz and feldspar) than roofing slates. But SPT sample is wholly made up by phyllosilicates.

We correct the misunderstanding in this phrase, please check rows 209-211

Rows 216 – “The skewness analysis (Figure 6) indicated that…….” you cited Figure 6B therefore I suggest shifting it in the left part of Figure 6 rename as Figure 6A.

Done

Rows 219 – 221. Please cite Figure 6A in the text now renamed as 6B.

Done

Rows 226-227 – “Distilled water exhibits higher values (an average of 4.5 cm parallel to lineation and 5.0 cm perpendicular to lineation), while Na2SO4 and NaCl have averages of 3.8 and 4.8 cm parallel to lineation, and 4.3 and 4.6 cm perpendicular to lineation, respectively.” You inverted some values please check accurately. I think that you also inverted the values in Table 2. I understand the symbols Wt but it is better to specify in table caption. Table 2 could be reorganized in the same order in which you have discussed the data in the text shifting the mineralogy on the left part of the table, then the surface roughness values and finally on the right part of Table 2, the capillary ascension data.

Values were inverted in the text, and the table has also some mistakes regarding the columns. We have corrected that, apologize for the errors.

Abbreviation Wt specified in the table caption

Table 2 reorganized as suggested

Row 250 – In Figure 6A and B substitute the colour of symbol NaCl with grey as indicated in Figure A and B.

Done

Figure 5 – It is better to insert a scale bar near the macro images.

Done. We also have included a scale bar in Figure 7.

Refences – Check accurately and where possible, use Journal title abbreviations.

References checked. We have used the MPDI style for Endnote provided by the publisher.

Reviewer 2 Report

In this paper, 24 has measured surface roughness in different types of roofing slates using a laser scanner and 25 determined the capillarity values along and across the grain direction.. This is a good study and its publication will be beneficial for scientific community and will add to the existing knowledge of the roofing slate; capillarity; salts; roof installation; surface roughness. This paper can be accepted for publication, if the following recommendations are incorporated:

Conclusions:
*    Should be revised by including some findings with numeric results, not just explaining in general. 
References:
*    Include recent and relevant references in the reference list

Author Response

Conclusions:
*    Should be revised by including some findings with numeric results, not just explaining in general. 

We have included some numeric data in the conclusions

References:
*    Include recent and relevant references in the reference list

We have included recent references:

  • Wagner, H.W.; Jung, D.; Wagner, J.F.; Wagner, M.P. Slatecalculation—A practical tool for deriving norm minerals in the lowest-grade metamorphic pelites and roof slates. Minerals 2020, 10.
  • Perini, K.; Castellari, P.; Giachetta, A.; Turcato, C.; Roccotiello, E. Experiencing innovative biomaterials for buildings: Potentialities of mosses. Building and Environment 2020, 172, 8, doi:10.1016/j.buildenv.2020.106708.
